# Investigating the relationship between high-dose norepinephrine administration and the incidence of delayed cerebral infarction in patients with aneurysmal subarachnoid hemorrhage: A single-center retrospective evaluation

Andrea Cattaneo[1][☯], Christoph Wipplinger[1][☯], Caroline Geske[1], Florian Semmler[1], Tamara M. Wipplinger[1,2], Christoph J. Griessenauer[3,4], Judith Weiland[1], Alexandra Beez[1], Ralf-Ingo Ernestus[1], Thomas Westermaier[5], Ekkehard Kunze[1], Christian Stetter[1]*

1 Department of Neurosurgery, University Hospital Würzburg, Würzburg, Germany, 2 Department of Biobehavioral Sciences, Teachers College, Columbia University, New York, NY, United States of America, 3 Department of Neurosurgery, Christian Doppler Klinik, Paracelsus Medical University, Salzburg, Austria, 4 Research Institute of Neurointervention, Paracelsus Medical University, Salzburg, Austria, 5 Department of Neurosurgery, Helios-Amper Klinikum Dachau, Dachau, Germany

☯ These authors contributed equally to this work.
* stetter_c@ukw.de

## Abstract

### Background

One of the longest-standing treatments to prevent delayed cerebral infarction (DCI) in patients with aneurysmal subarachnoid hemorrhage (aSAH) remains raising the blood pressure to a certain level of mean arterial pressure. This may require high doses of norepinephrine, which has been associated with severe end organ damage. With this study, we aimed to investigate the effects of norepinephrine on the incidence of DCI in a clinical setting.

### Methods

We conducted a retrospective evaluation of patients with aSAH admitted to our institution between November 2018 and March 2021. Potential risk factors for DCI were analyzed and significant predictors were assessed by means of a logistic regression analysis to account for potential confounders.

### Results

In this study, 104 patients were included. Hereof, 39 (38%) showed radiologic signs of DCI between day three and 14 post-intervention. These patients had more frequent vasospasms (n = 37 vs. 30, p = 0.022), a higher Hunt & Hess score (3 ± 2 vs. 2 ± 1, p = 0.004), a lower initial Glasgow Coma Scale score (9 ± 5 vs. 12 ± 4, p = 0.003) and received a higher median

**Data Availability Statement:** All relevant data are within the paper and its Supporting Information files.

**Funding:** The authors received no specific funding for this work.

**Competing interests:** The authors have declared that no competing interests exist.

norepinephrine dose (20,356μg vs. 6,508μg, p < 0.001). A logistic regression analysis revealed that only high-dose norepinephrine administration (OR 2.84, CI 1.56–7.8) and vasospasm (OR 3.07, CI 1.2–7.84) appeared to be significant independent risk factors for DCI.

## Conclusion

Our results indicate a significant association between higher dose norepinephrine administration and the occurrence of DCI. Future research including greater sample sizes and a prospective setting will be necessary to further investigate the relationship.

## Introduction

Aneurysmal subarachnoid hemorrhage (aSAH) represents a serious condition with high morbidity and mortality. Even if the ruptured aneurysm is treated sufficiently and in a timely manner, the risk of lethal complications remains elevated in the early phase after hemorrhage. Despite major advances in neurointensive care and neurorehabilitation, patients are at high risk for significant disability. Half of all survivors cannot return to their previous occupation, and up to a quarter rely on others for care [1]. Delayed cerebral infarction (DCI) is among the most common complications during the first two weeks following the initial hemorrhage [2]. While extensive research on potential causes of DCI in aSAH patients has been conducted, the underlying mechanism remains incompletely understood [3]. DCI used to be attributed to cerebral vasospasm, however, it may also occur independent of vasospasm [2]. More recently, the explanation of the mechanism underlying DCI include a chain of events triggered at macroscopic and microscopic levels that cause hyperacute, acute, and long-term damage. The acute phase after aSAH is characterized by an increase in intracranial pressure, a reduction in cerebral blood flow, and consequently a reduction in cerebral perfusion. In case of severe ischemia, massive infarction develops, leading to fatal herniation. If the bleeding stops and intracranial hypertension declines, cerebral perfusion pressure normalizes. However, there is evidence that cerebral perfusion remains severely impacted due to microcirculatory dysfunction in brain parenchyma. The subacute phase usually lasts up to 72 hours. Here, disturbances of brain tissue perfusion are caused by a variety of mechanisms such as inflammatory changes, spreading depolarizations, apoptosis, disruption of the blood-brain barrier, microthrombosis, or microvasospasm formation, which result in tissue ischemia, brain edema formation, and ultimately cerebral infarction [1, 4–9].

Despite progress in understanding the pathophysiological mechanism that causes DCI, there have not been major changes in therapy over the past decade. While traditionally treated with triple H therapy (i.e., hypervolemia, hemodilution and hypertension), more recent guidelines recommend euvolemia and mild hypertension as a prophylactic treatment for vasospasm and DCI [10]. However, especially in intubated patients under general intravenous anesthesia, maintaining a certain degree of arterial hypertension may require high doses of intravenous catecholamines. It is well known that high-dose catecholamine therapy can induce limb necrosis or end organ damage due to the extensive constriction of small peripheral blood vessels [11, 12]. Yet, literature on the effects on cerebral microvasculature is scarce. Only few experimental animal studies have hinted that norepinephrine may cause vasoconstriction in cerebral microvasculature [13–15]. Moreover, an animal model has shown that brain vessels are particularly responsive to norepinephrine after aSAH [15]. We conducted a retrospective evaluation in patients with aSAH admitted to our institution to evaluate the relationship of catecholamines administered with the incidence of DCI in a clinical setting.

## Materials and methods

### Population and study design

Records from aSAH patients who were admitted to the authors' department between November 2018 and March 2021 were retrospectively reviewed and analyzed. The study was conducted in accordance with the Declaration of Helsinki and approved by the authors' department's ethics board.

We included patients aged 18 or older with an aSAH caused by one or more aneurysms, when records were complete. Patients with aSAH from any causes other than an intracranial aneurysm were excluded from this analysis. We also excluded patients with incomplete records on catecholamine therapy as well as patients treated with more than one catecholamine or with any catecholamines other than norepinephrine. Additionally, patients who spent less than 14 days in our intensive care unit (ICU) for any reason other than demise were excluded from the analysis.

All patients were monitored in our ICU for at least 14 days after hemorrhage and treated according to the authors' institutional standards as described below. Patient data was recorded for the first 14 days after aSAH.

### Institutional standard management of aSAH patients

The study was conducted in a university hospital with a high volume and a long-standing expertise in the management of aSAH patients. In our institution, patients with aSAH are treated in a neurosurgical ICU led by board certified neurosurgeons with a subspecialty certificate in intensive care medicine, in close collaboration with anesthesiologists for more complex ventilation associated issues.

Upon admission, all patients receive a cranial computed tomography (CT) scan. Patients are assessed by the Hunt & Hess (H&H) and Glasgow Coma Scale (GCS) scores. If patients present with a GCS score $\leq 8$, they are intubated for airway protection. If intubated before admission, the prior clinical status is recorded. All patients receive a central venous line and invasive blood pressure monitoring and fluid balance is recorded continuously throughout the ICU course.

An external ventricular drainage is implanted if GCS score is $\leq 8$ and CT scans suggested elevated intracranial pressure, edema, or hydrocephalus.

Whenever possible, digital subtraction angiography (DSA) and early aneurysm treatment, either surgical or endovascular, are performed within 24 hours after initial presentation.

After aneurysm treatment, euvolemia with a target hemoglobin value $\geq 10$ g/dl and a target central venous pressure with 10-12mmHg is maintained. The aim for a mean arterial pressure (MAP) is approximately 90mmHg and $< 160$mmHg for systolic blood pressure. Target values are maintained using isotonic fluid infusion, packed red blood cells, and norepinephrine. All patients receive continuous intravenous magnesium for neuroprotection, vasospasm, and DCI prophylaxis with a target serum concentration of 2–2.5 mmol/L for the first 14 days following aSAH [16–18].

After aneurysm treatment, a weaning attempt is made. If it fails, patients are kept under continuous intravenous anesthesia with midazolam and sufentanil.

Then, intracranial pressure monitoring is performed via the external ventricular drainage (EVD). In case of EVD exhaustion and/or uncertain values, an additional intraparenchymal probe is implanted for continuous monitoring. For vasospasm monitoring, daily transcranial doppler sonography (TCD) is performed. Vasospasm on TCD is defined as mean flow velocity over 140 cm/sec in the anterior circulation and 90 cm/sec in the basilar artery, or an increase

of more than 30 cm/s within 24 hours. If a vasospasm is suspected based on clinical symptoms (i.e., clinical vasospasm) or TCD findings, a CT-perfusion map or magnetic resonance imaging (MRI) including diffusion weighted images and perfusion maps is obtained.

If a patient is symptomatic or perfusion deficits on CT or MRI perfusion maps are seen, a DSA is performed under general anesthesia with the aim of vasospasmolysis, either by intraarterial nimodipine infusion, balloon angioplasty, or a hybrid technique, if necessary.

All patients are monitored on the ICU until their discharge into a rehabilitation facility.

## Imaging analysis

If awake patients are capable of neurological examination, CT or MRI scans are performed in case of clear or suspected clinical deterioration, or if EVD problems require CT scans. In patients under continuous sedation, routine CT scans are obtained on days three or four, six or seven, nine or ten post-aSAH, and before discharge, or when considered necessary for therapy.

Hypodensities were classified as: (1) preexisting; (2) exclusively resulting from intracerebral hematoma; (3) caused by operative procedures; or (4) DCI defined as hypodensities on CT or respective findings in MRI appearing between day three and the end of the observation period after exclusion of procedure related infarctions [19]. Angiographic vasospasm was defined as narrowing of the arterial diameter of > 30% in DSA with significant delay of circulation time.

## Variables and measurements

The primary outcome variable for our study was DCI, defined as above and proposed by Vergouwen et al. [19]. To account for factors contributing to the development of DCI, we recorded patient demographics and preexisting cardiovascular risk factors including gender, age, smoking status, diabetes, obesity (i.e., BMI > 30) and arterial hypertension. Additionally, we evaluated the GCS on admission as well as the aSAH severity assessed by the H&H score. The location of the aneurysm identified as the most likely bleeding source was separated in two groups, anterior and posterior circulation. The aSAH pattern was graded according to the modified Fisher Scale.

Vasospasm was only recorded if it was confirmed on DSA according to the abovementioned definition. Moreover, we recorded interventions for vasospasm (i.e., intraarterial administration of nimodipine or balloon angioplasty).

Norepinephrine administration over the first 14 days following aSAH was recorded with the cumulative 14-day dose and the average 14-day dose per minute per patient expressed in micrograms (μg). We only included information on norepinephrine given during ICU stay. We did not record norepinephrine administration during angiography for vasospasm, or surgery, due to intraarterial nimodipine therapy or catecholaminergic support in case of bleeding or anesthesiologic complications. Additionally, we recorded the average MAP and heart rate per patient over the first 14 days after admission.

The functional outcome at discharge was defined using the modified Rankin scale (mRS).

## Statistical analysis and data collection

Data was extracted from our institutional general patient data management system SAP (SAP AG, Wallendorf, Germany) as well as from our ICU patient data management system COPRA (COPRA System Corporation, Berlin, Germany).

The Kolgomoronov-Smirnov test was used to determine normal distribution. Normally distributed data were expressed as mean ± standard deviation and skewed data as median and interquartile range (IQR) with the $25^{th}$ and $75^{th}$ percentiles.

Relationships between categorical variables were determined by the Chi-Square test. The Mann-Whitney-U was used to compare differences between continuous and nominal variables. A p-value < 0.05 was considered statistically significant and all p-values were two-tailed. Significant predictors for DCI from the univariate analysis were included in a logistic regression analysis to account for variable interactions. The regression model was first performed with continuous variables. As next step, the variables were converted into binary group variables to better express the magnitude of the effect the variables had on the occurrence of DCI. The cumulative 14-day norepinephrine dose was grouped into above and below the median, H&H into ≤ 3 and 4–5, GCS in ≤ 8 and 9–15 and modified Fisher Scale into < 4 and 4. Then, the regression model was repeated with the grouped variables and results expressed as odds ratio (OR) and 95% confidence interval. Additionally, the abovementioned analysis was repeated to identify significant predictors for poor functional outcome defined as mRS 4–6 at discharge.

All statistical evaluations were performed with SPSS Version 26.0 (IBM Corp. Released 2012. IBM SPSS Statistics for Mac OS X, Version 21.0, NY: IBM Corp.).

## Results

### Patients characteristic

We identified 256 patients who were treated for aSAH in our department between November 2018 and March 2021. We excluded 64 patients because of non-aneurysmal causes for aSAH (i.e., perimesencephalic hemorrhage, traumatic, arteriovenous malformations), 23 patients were excluded because they already had a history of aSAH and suffered from a repeated hemorrhage, or were admitted for repeated treatment of their aneurysm, 21 patients were excluded because they stayed in our ICU for less than 14 days, and 44 patients were excluded due to incomplete records.

Ultimately, we included a total of 104 patients in this retrospective analysis. The age ranged from 22 to 81 years and the mean age in our sample was 59 (± 12) years.

In our study sample, 71% (n = 74) patients were female and 29% (n = 30) male. Hereof, 17 (16%) patients had a history of smoking, five (5%) had diabetes, six (6%) suffered from obesity, and 59 (57%) patients had a history of arterial hypertension. Patients presented at admission with a mean GCS score of 10 (± 5). Thirty-nine patients (38%) had a GCS score of less than 8. Among the included patients, 62 (60%) presented with a mild aSAH (H&H 1–3) while 42 (40%) had a severe aSAH (H&H 4–5). Mean value of the modified Fischer score on the initial cranial CT scan was 3 (± 1). Most of the patients (n = 79, 76%) received an EVD within the first 24 hours of hospitalization.

Patients were hospitalized in our institution for an average of 22 days (± 11). A total of 44 (42%) patients were ventilated over the entire observation period of 14 days and received a surgical tracheostomy between day seven and 14.

On average, 1.3 (± 0.92) aneurysms per patient were found, and 153 aneurysms were diagnosed in the entire cohort. Hereof, 106 (69%) aneurysms were in the anterior circulation and 47 (31%) in the posterior circulation. The location and treatment modality of all aneurysms can be found in **Table 1**.

The aneurysm treatment was endovascular in 71 (68%), and microsurgical in 30 patients (29%), respectively. The remaining three (3%) patients were treated conservatively (i.e., without intervention) during our observation period. The reason for the conservative treatment was terminal prognosis in two patients (2%), and one (1%) treatment was delayed because of the complex configuration of the aneurysm requiring flow-diverter treatment and subsequent dual-anti-platelet therapy.

**Table 1. Baseline characteristics of the entire cohort consisting of 104 patients.**

| Baseline characteristics | | |
|---|---|---|
| | **n = or mean** | **% or SD** |
| Male (n and %) | 30 | 29% |
| Female (n and %) | 74 | 71% |
| Age at event in years (mean; SD) | 59 | ± 12 |
| Comorbidities | | |
| Smoker (n and %) | 17 | 16% |
| Diabetes (n and %) | 5 | 5% |
| Adipositas (n and %) | 6 | 6% |
| Hypertension (n and %) | 59 | 57% |
| Grading | | |
| GCS (mean; SD) | 10 | ± 5 |
| Mild aSAH (H&H 1–3) | 62 | 60% |
| Severe aSAH (H&H 4–5) | 42 | 40% |
| Modified Fisher Scale (mean; SD) | 3 | ± 1 |
| Aneurysm Location (n = 153) | | |
| Anterior Circulation (n and %) | 106 | 69% |
| Posterior Circulation (n and %) | 47 | 31% |
| Ruptured aneurysms (n = 104) | | |
| Anterior Circulation (n and %) | 68 | 70% |
| Posterior Circulation (n and %) | 36 | 30% |
| Aneurysm treatment | | |
| Endovascular (n and %) | 71 | 68% |
| Surgical (n and %) | 30 | 29% |
| No treatment (n and %) | 3 | 3% |
| EVD (n and %) | 79 | 76% |

n = number, SD = standard deviation, aSAH = aneurysmal subarachnoid hemorrhage, H&H = Hunt & Hess, GCS = Glasgow Coma Scale, EVD = external ventricular drain

## Vasospasm

In our cohort, 67 (64%) patients experienced at least one episode of vasospasm of which 23 (22%) patients developed clinically symptomatic vasospasm, and in 44 patients (42%) vasospasm was detected through TCD. Endovascular intervention for vasospasm was performed in 58 patients (56%), with 139 interventional procedures. In 117 (84%) cases, a chemical spasmolysis with nimodipine was performed, and in 22 (15%) cases, a balloon angioplasty was required. Among the patients with vasospasm, 2.4 (± 1.6) interventions per patient were performed on average. For details on vasospasm and subsequent treatments, see **Table 2**.

## Norepinephrine

Patients included in the study received a median cumulative norepinephrine dose of 14,224.50μg (2,285.5–31,680μg), with a median 14-day per minute rate of 0.71μg (0.12–1.57μg) during the first 14 days of treatment. During this time, the overall median of norepinephrine therapy was 7 days (0–14). Patients had a MAP of 94mmHg (± 7) and a mean heart rate of 69 BPM (± 15).

## Comparison between patients with DCI and patients without DCI

To determine risk factors associated with DCI, we divided our patients into two groups: DCI (n = 39, 37.5%) and no DCI (n = 65, 62.5%).

**Table 2. Total occurrence and type of intervention for vasospasm.**

| Vasospasm | | |
|---|---|---|
| | n = or mean | % or SD |
| Vasospasm (n and %) | 67 | 64% |
| Patients undergoing intervention for vasospasm | 58 | 56% |
| **Total interventions for vasospasm (n and %)** | **139** | |
| Nimodipine (n and %) | 117 | 84% |
| Angioplasty (n and %) | 22 | 15% |
| Average number of interventions per patient (mean; SD) | 2.4 | ± 1.60 |

n = number, SD = standard deviation

Patients in the DCI group had a significantly worse average initial GCS (9 vs. 12, p = 0.003) and H&H score (3 vs. 2, p = 0.004). Moreover, patients with DCI appeared to have more severe bleeding demonstrated by a worse modified Fischer Scale (4 vs. 3, p = 0.015) on the initial CT scan. Patients in the DCI group developed significantly more frequent vasospasm than those in the no DCI group (37 vs. 30, p = 0.022). Also, patients in the DCI group were significantly longer ventilated than patients without DCI (12 ± 6 vs. 6 ± 6 days, p = 0.004).

Regarding norepinephrine, we found significantly higher cumulative doses in patients with DCI. Patients with DCI received a median norepinephrine dose of 20,356μg (11,654–57,496μg) during the first 14 days of treatment, while a median dose of 6,508μg (0–12,862μg) was administered in patients who were not diagnosed with a DCI (p < 0.001). Moreover, patients with DCI received norepinephrine for 11 days (3–14), patients without DCI 3 days (0–8), which is significantly shorter (p = 0.005).

The MAP was similar in both groups (92 vs. 95, p = 0.251). In **Table 3** a detailed list of the two groups is provided.

To account for confounding bias, variables that showed a significant association with DCI were included in a logistic regression model as covariates with DCI as dependent variable. We found that only the presence of vasospasm (p = 0.031) and the cumulative 14d norepinephrine dose (p = 0.018) retained a statistically significant association with the occurrence of DCI. To obtain an OR, the regression analysis was repeated with the variables in binary groups as described in the methods section. With norepinephrine doses above the sample's overall median (14,224.5μg), the OR for DCI was 2.84 (1.56–7.8) and the presence of vasospasm had an OR of 3.07 (1.20–7.84) for DCI. A detailed list of the group analysis can be found in **Table 4**.

### Clinical outcome

We performed a univariate analysis to find variables with significant association with poor outcome.

After accounting for confounding variable interaction in a logistic regression analysis, only the presence of DCI appeared to be a significant predictor for poor outcome with an OR 9.35 (2.26–38.71, p = 0.003). The detailed analysis can be seen in **Table 5**.

## Discussion

### Risk factors for DCI

We identified a cumulative 14-day norepinephrine dose over 14,224.5μg and angiographic vasospasm as independent risk factors for DCI defined as radiologic evidence of cerebral infarction occurring in a delayed fashion at least three days after aSAH [19].

Previous studies suggest that approximately 30–40% of aSAH patients develop DCI [20, 21]. The pathophysiologic mechanisms contributing to the development of DCI are still not

**Table 3. Potential risk factors for DCI.** Relationships between categorical variables. Mann-Whitney-U was used for the comparison of DI and continuous variables.

| | Patients with DCI (n = 39, 37.5%) | % or SD or IQR | Patients without DCI (n = 65, 62.5%) | % or SD or IQR | P Value |
|---|---|---|---|---|---|
| Male (n and %) | 7 | 7% | 23 | 22% | 0.074 |
| Female (n and %) | 32 | 31% | 42 | 40% | |
| Age at event in years (mean; SD) | 59 | ± 11 | 60 | ± 12 | 0.678 |
| **Comorbidities** | | | | | |
| Active Smoker (n and %) | 8 | 8% | 9 | 9% | 0.781 |
| Diabetes (n and %) | 2 | 2% | 3 | 3% | 0.495 |
| Adipositas (n and %) | 3 | 3% | 3 | 3% | 0.807 |
| Hypertension (n and %) | 31 | 30% | 28 | 27% | 0.595 |
| **Scores** | | | | | |
| GCS (mean; SD) | 9 | ± 5 | 12 | ± 4 | 0.003* |
| Hunt & Hess (mean; SD) | 3 | ± 2 | 2 | ± 1 | 0.004* |
| Modified Fisher Scale (n; SD) | 4 | ± 1 | 3 | ± 1 | 0.015* |
| **Treatment** | | | | | |
| EVD (n and %) | 33 | 32% | 46 | 44% | 0.52 |
| Ventilation days | 12 | ± 6 | 6 | ± 6 | 0.044* |
| **Aneurysm treatment** | | | | | |
| Endovascular (n and %) | 22 | 21% | 52 | 50% | |
| Surgical (n and %) | 16 | 15% | 11 | 7% | 0.007* |
| No treatment (n and %) | 1 | 1% | 2 | 2% | |
| **Bleeding source** | | | | | |
| Anterior circulation (n and %) | 33 | 32% | 46 | 44% | 0.155 |
| Posterior circulation (n and %) | 6 | 6% | 19 | 18% | |
| **Vasospasm** | | | | | |
| Vasospasm (n and %) | 37 | 36% | 30 | 29% | 0.022* |
| **Hemodynamic therapy** | | | | | |
| 14d cumulative norepinephrine dose in μg (median; IQR) | 20,356 | 11,654–57,496 | 6,508 | 0–12,862 | < 0.001* |
| 14d norepinephrine dose/min in μg (median; IQR) | 1.01 | 0.54–2.86 | 0.58 | 0–0.82 | < 0.001* |
| Days norepinephrine administered (median; IQR) | 11 | (3–14) | 3 | (0–8) | 0.005* |
| 14d MAP (mean; SD) | 92 | ± 7 | 95 | ± 6 | 0.241 |
| 14d HR (mean; SD) | 70 | ± 14 | 70 | ± 16 | 0.722 |

DCI = delayed cerebral ischemia, n = number, SD = standard deviation, IQR = interquartile range, GCS = Glasgow Coma Scale, EVD = external ventricular drain, d = days, μg = microgram, MAP = mean arterial pressure, HR = heart rate. *statistically significant

fully understood. While large vessel vasospasm is a well-studied risk factor for DCI [22], more recent studies have suggested that a considerable number of patients develop DCI without vasospasm [2, 3, 23]. Possible underlying pathomechanisms independent of large vessel vasospasm that have been proposed include microthromboembolisms, microvasospasms, inflammation, neurovascular mismatch and cortical spreading depolarization [1, 4, 24–27].

## Vasopressor therapy after intracranial hemorrhage

It is well known that excessive use of vasopressors comes with the risk of severe end organ damage, presumably caused by arteriolar and capillary vasoconstriction as well as peaks of systolic arterial blood pressure [28–30].

**Table 4. Potential risk factors for DCI.** Results from the logistic regression analysis expressed in odds ratio for the occurrence of DCI.

|  | OR (95% CI) | P value |
|---|---|---|
| Norepinephrine (cumulative 14d) > 14,224.5μg | 2.84 (1.56–7.8)* | 0.016* |
| Modified Fisher Scale 4 | 1.81 (0.22–3.23) | 0.236 |
| Hunt & Hess Grade 4–5 | 2.72 (0.47–31.62) | 0.256 |
| Ventilation 14d | 3.15 (0.56–28.04) | 0.269 |
| GCS ≤ 8 | 2.31 (0.315–16.92) | 0.347 |
| Vasospasm | 3.07 (1.20–7.84)* | 0.039* |
| Endovascular surgical treatment | 1.92 (0.489–7.71) | 0.582 |

OR = odds ratio, CI = 95% confidence interval, d = days, μg = microgram, GCS = Glasgow Coma Scale,
DCI = delayed cerebral ischemia.
*statistically significant

Current guidelines do not provide specific recommendations on arterial blood pressure targets for prophylactic or therapeutic induced hypertension but in general, the majority of the literature reports prophylactic MAP goals between 80 and 130mmHg [22, 31, 32].

In our institution, we almost exclusively use norepinephrine except for specific circumstances such as congestive heart failure or cardiac insufficiency.

To date, there is considerable controversy which vasopressor should be utilized in patients with cranial pathologies. Currently, the most commonly used vasopressors are phenylephrine, norepinephrine, and dopamine [29, 33]. There is a certain consensus that dopamine is associated with hyperemia, increased risk of brain edema, and subsequent intracranial pressure elevation, and should therefore be avoided in patients with intracranial hemorrhage [34–36]. A recent nationwide retrospective cohort study from the United States compared clinical outcomes and complications of all three vasopressors in aSAH patients and found significantly superior outcomes and fewer systemic complications in patients treated with phenylephrine [29]. This contradicted the findings of an earlier study that compared phenylephrine and norepinephrine in aSAH patients and found better clinical outcomes in patients treated with norepinephrine. The study was, however, rather focused on direct comparison of different

**Table 5. Potential risk factors for poor outcome.** Results from the univariate anaylsis expressed with a p-value as well as results from the subsequent logistic regression analysis expressed in odds ratio and p-value.

| | Outcome at discharge | | | | |
|---|---|---|---|---|---|
| | good (mRS 0–3) n = 32 | poor (mRS 4–6) n = 72 | Univariate analysis | Logistic regression analysis | |
| | | | P Value | OR (95% CI) | P Value |
| Vasospasm | 18 | 49 | 0.147 | 2.02 (0.567–7.22) | 0.105 |
| DCI | 5 | 34 | <0.001* | 9.35 (2.26–38.71) | 0.003* |
| H&H 4–5 | 2 | 35 | <0.001* | 3.66 (0.369–36.24) | 0.079 |
| Norepinephrine (cumulative 14d) > 14,224.5μg | 7 | 45 | <0.001* | 2.85 (0.82–9.98) | 0.056 |
| GCS ≤ 8 | 3 | 36 | <0.001* | 6.50 (0.87–48.56) | 0.347 |
| Modified Fisher Scale 4 | 6 | 28 | 0.044* | 1.54 (0.31–7.61) | 0.228 |
| Surgical treatment | 6 | 26 | 0.169 | 1.19 (0.95–1.5) | 0.195 |

mRS = modified Rankin Scale, OR = odds ratio, CI = confidence interval, DCI = delayed cerebral ischemia, H&H = Hunt & Hess, d = days, μg = microgram,
GCS = Glasgow Coma Scale
*statistically significant

vasopressors than on specific dose related adverse effects of norepinephrine and the development of DCI. [37]. Similarly, a review investigating the effects of dopamine, phenylephrine, and norepinephrine in patients with traumatic brain injury suggested that norepinephrine is the most predictable and therefore best suited vasopressor to maintain a certain cerebral perfusion pressure. However, a direct translation of findings in patients with traumatic brain injury can only be made to a limited extent since the pathophysiology of secondary brain damage in aSAH patients is inherently different [36].

## Cerebrovascular adverse effects of norepinephrine

In the present study, the cumulative catecholamine dose administered during the first two weeks after aSAH showed a significant association with the presence of DCI. While the median 14 day cumulative dose of norepinephrine was significantly higher in the DCI group, there have been patients with comparably high doses in the group non-DCI group as is reflected by the interquartile range. Since it is known that the development of DCI is multifactorial, we assume that there may be patients who tolerate higher doses of norepinephrine without developing DCI. Statistically, however, even after adjustment for potential confounding variables, our findings strongly indicate that higher doses of norepinephrine contribute to the development of DCI.

To date, literature indicating a relationship between high-dose norepinephrine and the development of DCI is scarce. The reason for the rare reporting of adverse effects of norepinephrine on DCI may originate from the fact that many studies report starting to administer vasopressors after a delayed ischemic neurologic deficit or even DCI occurs [29, 38, 39]. However, patients admitted to our institution routinely receive norepinephrine for MAP maintenance right after successful aneurysm treatment (i.e., within 24 hours), which inherently minimizes this potential bias. One of the few articles suggesting aggravation of DCI due to extensive norepinephrine administration is a small case series from 2014. The authors reported two cases of aSAH patients who showed clinical signs of DCI that rapidly deteriorated under aggressive induced hypertension therapy (MAP 150) and improved shortly after discontinuation and administration of milrinone [40].

At this point, explanations for this phenomenon can only be hypothesized. The theory that the mechanism underlying DCI is probably not only related to macrovessel vasospasm is nowadays widely accepted. Certainly, a role in the development of DCI is played by microvessel vasospasm that is not visible on conventional angiography. The effect of catecholamines, and more generally of vasoactive substances in the brain is still poorly understood [20, 23, 41, 42]. There is some evidence from mostly animal studies and human autopsy studies that the alpha-adrenoreceptors of cerebral vessels are stimulated by low concentrations of norepinephrine leading to vasoconstriction. An even stronger effect is possible at high concentrations when gamma adrenoreceptors are stimulated [43, 44]. A photoacoustic microscopy study from 2013 showed that intravenous norepinephrine administration in an in vivo mouse model induces direct vasoconstriction with reduction of hemoglobin concentration and oxygen saturation in cerebral arterioles and veins [13].

Moreover, aSAH seems to be a condition that increases the response to norepinephrine und serotonin, thus enhancing the vasoconstrictive effect of these substances [15].

The data available in literature are mostly not recent, and the models used are typically ex-vivo animal models. Those models do not include the effects of microvessel innervation that carries a norepinephrine-mediated sympathetic modulation. An underestimated aspect when considering the therapeutic use of norepinephrine is precisely its dual effect in the brain, as a hormone and as a neurotransmitter. It is known that there is damage at the level of the blood-

brain barrier in patients with aSAH. This could favor the uncontrolled entry of norepinephrine into the subarachnoid space. The noradrenergic neuronal network has been shown to have a modulatory role in cortical blood flow distribution. A study from 2012 using a mouse model demonstrated that a direct stimulation of the noradrenergic pathway induces an increase in cortical norepinephrine release with subsequent vasoconstriction. Lesions in the noradrenergic pathway in the medulla oblongata seem to prevent vasospasm after aSAH [45, 46].

Another aspect described in the literature is the prognostic role of catecholamine levels in the cerebrospinal fluid after aSAH. Several studies found elevated catecholamine levels in the cerebrospinal fluid of patients with higher grade aSAH [47, 48]. Additionally, these studies found a correlation of elevated catecholamine levels in the cerebrospinal fluid with poor clinical outcome. Since there is evidence suggesting a disturbance in the blood-brain barrier as well as impaired cerebral autoregulation in aSAH patients [23, 41, 49], it may be plausible that this particular subset of patients can experience detrimental effects from excessive norepinephrine administration.

A final important aspect to consider is the role of adrenergic receptors. In cardiology, there is a growing body of literature directed towards describing the effects of gene polymorphisms of adrenergic receptors. Apparently, these polymorphisms may be partially responsible for coronary artery vasoconstriction and sudden cardiac death as well as blood pressure dysregulation [50, 51]. Hence, the data in the present literature indicate a vasoconstrictive effect of norepinephrine at the level of the cerebral microcirculation that is possibly also reinforced by the noradrenergic pathway. Our data appear to clinically confirm the findings on the effects of norepinephrine at the brain level in animal models. The groups were homogeneous in MAP, which corroborates the theory that dose-related effects probably act not only on systemic blood pressure but also at the neuroreceptor level. Certainly, there is a lack of in-vivo data to support this hypothesis in a more complex setting.

## Limitations

The retrospective nature of our study inherently limits its explanatory power. In addition, the relatively small sample size along with the lack of a control group makes the generalizability of our findings challenging. In our sample, there is a significantly higher number of female patients. While it is known that stress cardiomyopathy occurs predominantly in female aSAH patients [52], we did not find any documentation of major cardiac events among the included patients. However, the possibility of an increased need for norepinephrine due to latent, undiagnosed aSAH-associated cardiomyopathy cannot be ruled out with absolute certainty.

Since we could only include patients with continuous recordings of intravenous medication doses and vital signs, we were only able to include 104 patients in this analysis. Additionally, according to our internal standard of care management, we generally tend to keep higher grade aSAH patients deeply sedated for neuroprotection over the first 14 days post hemorrhage if the first early weaning attempt fails. Therefore, recurrent vasospasm may be missed if TCD fails to reveal increased flow velocities. Furthermore, the continuous intravenous anesthesia may increase the need for vasopressor therapy to maintain the target MAP of 90mmHg. When comparing ventilation days, we found that patients with DCI were ventilated significantly longer than patients without. However, after performing a multivariate analysis by using a logistic regression model, the difference in ventilation days did not retain statistical significance. While the rate of vasospasm was increased in the group of patients with DCI, the average MAP was similar in both groups. This can be explained by the fact that, according to our internal standards, the maximum MAP level is set at 95mmHg.

Since the current study is of retrospective nature and included a small sample size, a confounding bias cannot be ruled out with absolute certainty. A prospective randomized-controlled study would be needed to further investigate the relationship of norepinephrine and DCI in aSAH patients.

## Conclusion

In our patient sample, we observed an association between higher dose norepinephrine administration and the occurrence of delayed cerebral ischemia even after adjusting for potential confounding variables. Aggressive attempts to maintain a certain MAP should be cautiously weighed against the risk of adverse effects from high-dose norepinephrine administrations. Further prospective studies with larger sample sizes and control groups as well as in-vivo studies on the effects of norepinephrine will be required to validate these results and to determine an accurate therapeutic index for norepinephrine.

## Supporting information

**S1 Dataset.**
(XLSX)

## Author Contributions

**Conceptualization:** Andrea Cattaneo, Christoph Wipplinger, Judith Weiland, Alexandra Beez, Thomas Westermaier, Christian Stetter.

**Data curation:** Christoph Wipplinger, Caroline Geske, Florian Semmler, Tamara M. Wipplinger, Judith Weiland, Alexandra Beez.

**Formal analysis:** Andrea Cattaneo, Christoph Wipplinger, Tamara M. Wipplinger.

**Funding acquisition:** Christian Stetter.

**Methodology:** Andrea Cattaneo, Christoph J. Griessenauer.

**Project administration:** Tamara M. Wipplinger.

**Supervision:** Christoph Wipplinger, Christoph J. Griessenauer, Ralf-Ingo Ernestus, Ekkehard Kunze, Christian Stetter.

**Validation:** Christoph Wipplinger, Christian Stetter.

**Writing – original draft:** Andrea Cattaneo, Christoph Wipplinger, Tamara M. Wipplinger.

**Writing – review & editing:** Andrea Cattaneo, Christoph Wipplinger, Tamara M. Wipplinger, Christoph J. Griessenauer, Christian Stetter.

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
