## [Decision Letter · Decision Letter 0]

8 Jan 2023

PONE-D-22-33697Investigating the relationship between high-dose norepinephrine administration and the incidence of delayed cerebral infarction in patients with aneurysmal subarachnoid hemorrhage: A single-center retrospective evaluationPLOS ONE

Dear Dr. Wipplinger,

Thank you for submitting your manuscript to PLOS ONE. After careful consideration, we feel that it has merit but does not fully meet PLOS ONE’s publication criteria as it currently stands. Therefore, we invite you to submit a revised version of the manuscript that addresses the points raised during the review process.

We look forward to receiving your revised manuscript.

Kind regards,

Martin Kieninger

Academic Editor

PLOS ONE

Reviewers' comments:

Reviewer's Responses to Questions

**Comments to the Author**

1. Is the manuscript technically sound, and do the data support the conclusions?

Reviewer #1: Partly

Reviewer #2: Yes

2. Has the statistical analysis been performed appropriately and rigorously? 

Reviewer #1: Yes

Reviewer #2: No

3. Have the authors made all data underlying the findings in their manuscript fully available?

Reviewer #1: Yes

Reviewer #2: No

4. Is the manuscript presented in an intelligible fashion and written in standard English?

Reviewer #1: Yes

Reviewer #2: Yes

5. Review Comments to the Author

Reviewer #1: In the present paper Cattaneo et al investigated the relationship between high-dose norepinephrine

administration and the incidence of delayed cerebral infarction in patients with aneurysmal subarachnoid hemorrhage.

The paper is of major general interest since the treatment modalities for cerebral vasospasm following aSAH changed significantly over the years due to new findings. Induced hypertension using catecholamines remains the last part of the formerly used HHH therapy and even induced hypertension is actually discussed controversially. Thus this paper contributes to the information about the risks and benefits of this widely used treatment.

The paper is well written with only a few typos and punctuation errors (see attachment).

"Materials and Methods" is well written and sufficiently describes the base for the results.

"Results" part is clear and describes all important results. The only problem in my opinion is that there is a significantly higher number of female patients in the DCI group compared to male patients. This might be important since female patients seem to be more often prone to cardiac impairment following aSAH, thus potentially needing additional positive inotropy besides norepinephrine.

Also, to name the norepinephrine dosage high or excessive, it must be documented in µg/kg/min. Unfortunately, there is only the mean dosage for all patients in µg/kg/min documents and not the difference in the two studied groups. The higher cumulative dosage in the DCI group could be attributed to a longer time of administration and not to higher doses at specific time points. Please clarify if the excessive dosage of norepinephrine is only attributed to the dose over time. In addition, it would be interesting if in the DCI group are also patients with a comparable norepinephrine dosage to the non-DCI group. This would than also need an additional explanation.

Discussion:

The discussion is well-written and contains interesting and major points to support the authors' theory that norepinephrine might be an additional risk for DCI following SAH.

But there are some ambiguities. The authors discuss that there might be higher norepinephrine administration in the DCI group due to a higher rate of vasospasm but state that the difference of MAP between the groups was similar (95 mmHg vs 92 mmHg) due to the standards at their institution (opt for MAP between 90 and 95 mmHg). So there has to be another reason for the higher norepinephrine dosage in the DCI group to reach the target MAP, but there is no explanation found in the discussion (e.g. neurogenic myocardial injury (NMI)). If NMI occurred in the DCI patients more often than in the non-DCI patients it might be possible that a higher norepinephrine dosage might be detrimental if no positive inotropic substance is used in addition due to impaired cardiac ejection fraction. This might lead to impaired organ perfusion and oxygen delivery.

My last point is that the authors set a standard MAP for patients with severe vasospasm. At least a part of the documented DCI could be part of a MAP that was set too low to allow adequate cerebral perfusion even if the authors used intraarterial chemical vasospasmolysis in patients with severe cerebral vasospasm. Maybe they can elucidate their opinion about this since there are reports that neurological deficits improved significantly after elevating MAP in awake patients with symptomatically cerebral vasospasm (we also see this very often at our neurointensive care unit).

Reviewer #2: My comments are as follows

1. Please discuss why your results are different from References 36 and 37,

2. Please explain the difference between patients who had norepinephrine but did not have DCI.

3. Why norepinephrin is a risk factor for DCI, but with or without norepinephrine there is no difference in the prognosis at discharge

6. PLOS authors have the option to publish the peer review history of their article (what does this mean?). If published, this will include your full peer review and any attached files.

Reviewer #1: No

Reviewer #2: No

---

## [Author Response · Author response to Decision Letter 0]

5 Feb 2023

Reviewer #1: 

In the present paper Cattaneo et al investigated the relationship between high-dose norepinephrine

administration and the incidence of delayed cerebral infarction in patients with aneurysmal subarachnoid hemorrhage.

The paper is of major general interest since the treatment modalities for cerebral vasospasm following aSAH changed significantly over the years due to new findings. Induced hypertension using catecholamines remains the last part of the formerly used HHH therapy and even induced hypertension is actually discussed controversially. Thus this paper contributes to the information about the risks and benefits of this widely used treatment.

The paper is well written with only a few typos and punctuation errors (see attachment).

"Materials and Methods" is well written and sufficiently describes the base for the results.

"Results" part is clear and describes all important results. 

R: We thank the reviewer for the positive feedback. Furthermore, we appreciate the reviewer’s acknowledgement of the contribution to the current literature we are attempting by publishing these findings.

The only problem in my opinion is that there is a significantly higher number of female patients in the DCI group compared to male patients. This might be important since female patients seem to be more often prone to cardiac impairment following aSAH, thus potentially needing additional positive inotropy besides norepinephrine.

R: The reviewer makes an excellent point. The concern is valid and may present a potential bias. No other catecholamines apart from norepinephrine (e.g., dopamine) were administered in our present study sample. We did not find any documentation of major cardiac events among the included patients. However, the possibility of an increased need for norepinephrine due to latent, undiagnosed aSAH-associated cardiomyopathy cannot be ruled out with absolute certainty. A paragraph acknowledging this potential bias has been added to the discussion section. 

Also, to name the norepinephrine dosage high or excessive, it must be documented in µg/kg/min. Unfortunately, there is only the mean dosage for all patients in µg/kg/min documents and not the difference in the two studied groups. 

R: The µg/min doses for both groups (i.e., DCI and no DCI) are mentioned in Table 3. Since this information only reflects the average µg/min over 14 days we now additionally included the median number of days on which norepinephrine was actually administered in both groups (i.e., DCI and no DCI). 

The higher cumulative dosage in the DCI group could be attributed to a longer time of administration and not to higher doses at specific time points. 

Please clarify if the excessive dosage of norepinephrine is only attributed to the dose over time. 

R: We agree with the reviewer’s comment. Certainly, the amount of time patients are exposed to norepinephrine is a contributing factor. The average days of norepinephrine administration for both groups have been added to Table 3.

In addition, it would be interesting if in the DCI group are also patients with a comparable norepinephrine dosage to the non-DCI group. This would than also need an additional explanation.

R: While the median 14d cumulative dose of norepinephrine, the median µg/min dose as well as the total days norepinephrine was administered were higher in the DCI group, there have been patients with comparably high doses in the non-DCI group as is reflected by the IQR. It is known that the development of DCI is multifactorial. Therefore, we assume that there may be patients who tolerate higher doses of norepinephrine without developing DCI. Statistically, however, our findings indicate that higher doses of norepinephrine contribute to the development of DCI. We have added an according paragraph to the discussion in order to address this issue. 

Discussion:

The discussion is well-written and contains interesting and major points to support the authors' theory that norepinephrine might be an additional risk for DCI following SAH.

But there are some ambiguities. 

The authors discuss that there might be higher norepinephrine administration in the DCI group due to a higher rate of vasospasm but state that the difference of MAP between the groups was similar (95 mmHg vs 92 mmHg) due to the standards at their institution (opt for MAP between 90 and 95 mmHg). So there has to be another reason for the higher norepinephrine dosage in the DCI group to reach the target MAP, but there is no explanation found in the discussion (e.g. neurogenic myocardial injury (NMI)). 

If NMI occurred in the DCI patients more often than in the non-DCI patients it might be possible that a higher norepinephrine dosage might be detrimental if no positive inotropic substance is used in addition due to impaired cardiac ejection fraction. This might lead to impaired organ perfusion and oxygen delivery.

R: We appreciate this productive comment. As mentioned above, we did not find any documentation of major cardiac events among the included patients. However, latent cardiac insufficiency resulting in an increased need for catecholamines to maintain the target MAP may have gone unnoticed, but it is unlikely that it would be so frequent as to justify the significant difference in norepinephrine dosage. This has been added to the discussion. 

My last point is that the authors set a standard MAP for patients with severe vasospasm. At least a part of the documented DCI could be part of a MAP that was set too low to allow adequate cerebral perfusion even if the authors used intraarterial chemical vasospasmolysis in patients with severe cerebral vasospasm. Maybe they can elucidate their opinion about this since there are reports that neurological deficits improved significantly after elevating MAP in awake patients with symptomatically cerebral vasospasm (we also see this very often at our neurointensive care unit).

R: We thank the authors for this valuable comment. According to our department’s SOPs we avoid forcing an MAP beyond 95 due to concerns about systemic side effects of excessive induced hypertension. Instead, we rather perform endovascular vasospasmolysis more frequently in patients with refractory vasospasm. We can, however, not rule out the possibility that a certain subset of patients would have benefited from more extensive induced hypertension. Since a major part of the study sample was under deep sedation for most of the observation period (as reflected by the ventilation days in Table 3), we may have missed persistent ischemic neurological deficits if TCD did not reveal elevated flow velocity. A comment on this issue has been added to the discussion section. 

Reviewer #2: 

My comments are as follows

1. Please discuss why your results are different from References 36 and 37

R: We thank the reviewer for pointing out this issue. 

Reference Nr. 36 refers to a different subset of patients (patients with TBI) with a different pathophysiology, therefore we believe the comparability with aSAH patients is limited. As for reference Nr. 37, the article only compared the effects of different vasopressors on the development of DCI, and as described in this study, norepinephrine had the best outcome in direct comparison with other vasopressors. The study did, however, not investigate the dose-dependent effects of norepinephrine. Accordingly, a paragraph addressing this issue has been added to the discussion section. 

2. Please explain the difference between patients who had norepinephrine but did not have DCI.

R: The median 14d cumulative dose of norepinephrine as well as median µg/min dose, the peak dose and the total days on which norepinephrine was administered were higher in the DCI group. There have been patients with comparably high doses in the group non-DCI group as is reflected by the IQR. It is known that the development of DCI is multifactorial. Therefore, we assume that there may be patients who tolerate higher doses of norepinephrine without developing DCI. Statistically, however, our findings indicated that higher doses of norepinephrine may contribute to the development of DCI. To address this issue, we have added a paragraph to the discussion. 

3. Why norepinephrin is a risk factor for DCI, but with or without norepinephrine there is no difference in the prognosis at discharge

R: We apologize for not pointing this out more clearly. In the univariate analysis, norepinephrine was a significant risk factor for poor outcome. However, we then included norepinephrine and DCI in a logistic regression model along with other factors to minimize the risk confounding bias. In this analysis, only DCI retained its significance as independent risk factor for poor outcome. Since norepinephrine appears to contribute to the development of DCI, it has not retained statistical significance when measured along with DCI in a multivariate model. To better clarify significant risk factors for poor outcome, results of a univariate analysis have been added to Table 5.

---

## [Decision Letter · Decision Letter 1]

6 Mar 2023

Investigating the relationship between high-dose norepinephrine administration and the incidence of delayed cerebral infarction in patients with aneurysmal subarachnoid hemorrhage: A single-center retrospective evaluation

PONE-D-22-33697R1

Dear Dr. Wipplinger,

We’re pleased to inform you that your manuscript has been judged scientifically suitable for publication and will be formally accepted for publication once it meets all outstanding technical requirements.

Kind regards,

Martin Kieninger

Academic Editor

PLOS ONE

Additional Editor Comments (optional):

Reviewers' comments:

Reviewer's Responses to Questions

**Comments to the Author**

1. If the authors have adequately addressed your comments raised in a previous round of review and you feel that this manuscript is now acceptable for publication, you may indicate that here to bypass the “Comments to the Author” section, enter your conflict of interest statement in the “Confidential to Editor” section, and submit your "Accept" recommendation.

Reviewer #1: All comments have been addressed

Reviewer #2: All comments have been addressed

2. Is the manuscript technically sound, and do the data support the conclusions?

Reviewer #1: Yes

Reviewer #2: Yes

3. Has the statistical analysis been performed appropriately and rigorously? 

Reviewer #1: Yes

Reviewer #2: Yes

4. Have the authors made all data underlying the findings in their manuscript fully available?

Reviewer #1: Yes

Reviewer #2: Yes

5. Is the manuscript presented in an intelligible fashion and written in standard English?

Reviewer #1: Yes

Reviewer #2: Yes

6. Review Comments to the Author

Reviewer #1: Cattaneo et al. submitted the reviewed version of the paper "Investigating the relationship between high-dose norepinephrine administration and the incidence of delayed cerebral infarction in patients with aneurysmal subarachnoid hemorrhage: A single-center retrospective evaluation"

In its current form, the paper is suitable for publication since the authors addressed all my previous concerns.

I would add one sentence to elucidate/discuss how the higher incidence of vasospasm in the DCI group could influence the appearance of DCI independently from norepinephrine dosage. This would compliment to the quality of the paper in my opinion but is not needed for being accepted for publication.

Reviewer #2: The authors have replied my comments point by point. This revised manuscript got improved. I have no more comments. Accept is my fina ldecision

7. PLOS authors have the option to publish the peer review history of their article (what does this mean?). If published, this will include your full peer review and any attached files.

Reviewer #1: No

Reviewer #2: No

---

## [Editor Report · Acceptance letter]

13 Mar 2023

PONE-D-22-33697R1 

Investigating the relationship between high-dose norepinephrine administration and the incidence of delayed cerebral infarction in patients with aneurysmal subarachnoid hemorrhage: A single-center retrospective evaluation 

Dear Dr. Wipplinger:

I'm pleased to inform you that your manuscript has been deemed suitable for publication in PLOS ONE. Congratulations! Your manuscript is now with our production department. 

Kind regards, 

on behalf of

Dr. Martin Kieninger 

Academic Editor

PLOS ONE